# Amino Acids as Dietary Additives for Enhancing Fish Welfare in Aquaculture

**DOI:** 10.3390/ani15091293

**Published:** 2025-04-30

**Authors:** Natalia Salamanca, Marcelino Herrera, Elena de la Roca

**Affiliations:** 1Escuela Superior de Ingeniería, University of Huelva, 21071 Huelva, Spain; 2IFAPA Centro Agua del Pino, El Rompido-Punta Umbria rd., 21459 Cartaya, Spain; marcelino.herrera@juntadeandalucia.es

**Keywords:** amino acid, fish, animal welfare, aquaculture, nutrition

## Abstract

In aquaculture, fish are often exposed to stressors that affect survival or limit growth. In recent years, extensive research has been performed to reduce stress in fish in order to improve the welfare of farmed fish. The inclusion of beneficial additives in the diet in order to mitigate the stress response to typical stressors has been an important research topic. In addition, nutritional studies have shown that dietary supplementation with several amino acids (e.g., arginine, glutamine, glutamate, leucine, and proline) modulates this stress response. Nevertheless, it must be taken into account the high diversity of physiological effects depending on species and stressor type. This review summarizes the different studies on amino acid supplementation and its various attributes in fish culture and welfare.

## 1. Introduction

The interest in fish welfare in aquaculture facilities has grown significantly during the last decade. Important knowledge has been generated about fish stress, health, and pain; hence, currently, animal welfare is also a quality mark in aquaculture products, as in livestock species [1,2]. In fact, Wolke et al. [3] pointed out that fish farmers should recognize that the key to success is based on keeping fish stress-free. This statement is supported by fish farmers, who accept that healthy and well-being fish grow faster and better, resulting in economic benefits, and poor welfare is related to pathologies and immune failures [4]. For instance, reducing handling stress in cultured Senegalese sole (*Solea senegalensis*) resulted in lower cortisol levels and improved growth [5]. Stocking density is also a typical stressor in sea farms, though the optimal range is very variable depending on species and culture conditions [6,7,8]. The right management of this variable can improve fish welfare and enhance growth significantly [9].

There are many procedures for improving animal welfare in sea farms. Innovative zootechnical procedures are based on the improvement of the water quality through new culture systems like IMTAs (Integrated Multi-Trophic Aquaculture), organic cultures, bioflocs, etc. [10]. Stocking density handling is also an efficient strategy for enhancing fish welfare, though it is very variable depending on species and culture conditions [7,11,12,13]. Lastly, the enrichment of tanks seems to be a useful tool but maybe not very practical for intensive or semi-intensive systems [14,15].

Nevertheless, it is widely known that fish feeding and nutrition are key for keeping fish healthy [16]. Besides nutritional factors, many other food ingredients can play a crucial role in the physiological state and responses to stressful conditions. They are usually additives that are added to the conventional feed formulation, representing less than 5% of meal weight [17]. Many proteins, carbohydrates, fatty acids, vitamins, amino acids, and minerals have been studied as welfare promoters since they can modulate many metabolic pathways involved in the stress response [18].

Amino acids are a heterogeneous group with diverse biochemical functions; hence, they are involved in very different physiological processes. For instance, amino acids can be involved in the formation of neurotransmitters, part of energy metabolism pathways, or play immune functions [9,19]. Therefore, amino acids are maybe the organic molecules that have been the most studied in order to improve fish welfare through their inclusion in fish feeds. Amino acids are usually included in crystalline form. This supposes some limitations regarding their utilization efficiency because of lower rates of intestinal absorption and alteration of acid–base and electrolyte balance during digestion, among other causes [9,20]. However, this matter can be improved through a suitable feeding strategy [21].

In this review, the literature on the effects on dietary amino acids of fish stress and welfare has been revised. The objective is to show the state of the art in this area, analyzing and classifying the main current findings and future prospects in this field.

## 2. General Amino Acid Physiological Roles in Animals

Amino acids (AAs) are the structural units of proteins. AAs play a critical role in the organism as they perform important metabolic functions, regulating processes such as digestion, nutrient transport, and the production of enzymes and hormones. AAs are also essential for maintaining healthy immune systems.

There are more than 700 AAs in nature, but only 20 of them are building blocks for proteins and present in cells for the synthesis of polypeptides. These include alanine, arginine, asparagine, aspartate, cysteine, glutamate, glutamine, glycine, histidine, isoleucine, leucine, lysine, methionine, phenylalanine, proline, serine, tyrosine, threonine, tryptophan, and valine. However, selenocysteine and pyrrolysine are also considered amino acids, although they are not part of the standard amino acid set related to fish protein synthesis [22,23]. Ten of them are considered “essential” (EAA), meaning their carbon skeletons are not synthesized de novo or are not synthesized in adequate amounts by the organism. These essential AAs are arginine, histidine, isoleucine, leucine, lysine, methionine, phenylalanine, threonine, tryptophan, and valine. As a result, they must be provided through diets. On the other hand, the “nutritionally non-essential amino acids (NEAA)” can be synthesized within the body from carbon skeletons provided by cells or intermediates of metabolic pathways [24].

Amino acids act as substrates in different biosynthetic reactions. Furthermore, nucleotide bases and numerous hormones, such as catecholamines, serotonin, melatonin, and the thyroid hormones thyroxine and triiodothyronine, as well as neurotransmitters, are derived from amino acids [25,26]. Additionally, AAs are precursors for the synthesis of other important molecules such as glutathione and polyamines, all of which have diverse physiological functions. They can be synthesized from glycolytic or Krebs cycle intermediates [22,25].

The crucial roles of AAs in metabolism, physiology, and immunity against infectious diseases are particularly notable. Amino acids serve primary roles as precursors for protein synthesis, and they modulate the protein synthetic machinery within target cells [27]. From a physiological point of view, several amino acids have been shown to support the intestinal barrier function and the intestinal endocrine function [28]. Furthermore, adequate dietary provision of all amino acids is necessary for sustaining normal immunocompetence and protecting the host from a variety of diseases in all species [29]. In recent years, investigations have focused on regulatory roles for dietary AA in nutrient metabolism, including protein turnover, as well as lipid synthesis and oxidation, to favor lean tissue growth and adipose tissue reduction [30,31].

The role of amino acids in the immune system can also be considered from two perspectives: the enhancement of the immune response, which protects individuals from infections and malignant neoplasms, and the reduction of over-responses such as inflammation and autoimmunity [32].

Animals, including fish, need the same ten EAAs in their diet for proper development [21]. Similarly, the metabolic pathways of amino acids in fish are analogous to those of mammals. However, there are some variations, such as differences in the distribution of enzymes that deaminate branched-chain amino acids or in the secondary pathways of amino acid metabolism, such as those involving serine [33].

Fish and mammals exhibit key differences in amino acid metabolism, such as the limited ability of fish to synthesize certain amino acids and their lower efficiency in storing and converting proteins. Due to these differences, fish have a higher protein requirement than mammals, as they need less non-protein energy compared to homeotherms. In addition to the ten essential amino acids (EAAs) mentioned above, each species has distinct EAA requirements, influenced by factors such as their position in the food chain and water temperature. Among these EAAs, arginine, isoleucine, lysine, and threonine are particularly important in the fish diet, and these nutritional needs should be carefully considered in aquaculture to optimize fish welfare and productivity [21].

Nutritional studies have demonstrated that dietary supplementation with certain amino acids (referred to as functional amino acids) regulates key metabolic pathways to improve survival, development, growth, health, welfare, and reproduction of organisms. In this context, the most studied amino acids include arginine, glutamine, glutamate, tryptophan, sulfur amino acids (methionine, cysteine, and taurine), and histidine [34,35,36,37]. However, the works focusing on the interactions between stress responses and dietary amino acids in fish are based on other amino acids, mainly the essential ones, like tryptophan, taurine, and phenylalanine (Figure 1).

## 3. Stress Responses Depending on Amino Acids

Stress responses are diverse, including changes at different levels. Firstly, the endocrine and neuroendocrine changes are evident (primary response) and lead to alterations and failures in metabolism, immune system, reproduction, behavior, and growth [38]. Dietary amino acids can modulate those responses, though their effects are highly diverse depending on factors such as fish species, stressor type, and amino acid concentration. Table 1 summarizes the main physiological effects of amino acid additives on the stress response in fish, and that information has been developed below.

### 3.1. Immunological Responses

The immune response is closely linked to fish stress and welfare since stress leads to an endocrine cascade through the HPI (hypothalamic–pituitary–interrenal) axis, which affects the immune system [63]. The stressors trigger a series of regulatory events, as the hormonal activity increases, that interfere with the normal function of the immune system and facilitate a rapid pathogenic expansion [64]. For instance, post-stress blood cortisol increase can down-regulate the pro-inflammatory cytokine production [65] and depress phagocytosis of peripheral blood leukocytes [66].

It has been reported that some amino acids can modulate those stress immune effects. Tryptophan (Trp) is an essential amino acid and serotonin precursor, a neurotransmitter involved in stress response that can regulate the GALT (gut-associated lymphoid tissue) immune-related gene expression and anti-inflammatory signaling molecules [51,67,68]. It is hypothesized that stress modulation due to dietary Trp derives from the interaction between serotonergic activity and the HPI axis, though the physiological mechanisms regulating that interaction still remain unclear [44,69]. Additionally, Trp has also been shown to have significant impacts on immune functions controlled by specific pathways via its role as a structural component of specific transcription factors [46,70].

Herrera et al. [44] reported that Trp-enriched diets increased mucus antibacterial activity in the meager (*Argyrosomus regius*). Similarly, Gonzalez-Silvera et al. [47] analyzed several immunological parameters and concluded that dietary Trp improved the meager health by maintaining the levels of proteases and antiproteases under stress conditions. Lastly, Asencio-Alcudia et al. [46] concluded that Trp-supplemented diets can alleviate some of the most damaging aspects of the stress response on the mucosal immunity of the gut, such as elevated antimicrobial peptide expression that could change the microbiota and have long-term negative effects such as outbreaks of opportunistic bacterial pathogens.

Arginine (Arg) is a non-essential amino acid precursor for the synthesis of nitric oxide and polyamides [71], which is related to cellular defense mechanisms and can stimulate macrophages in fish [72,73]. Arg-enriched diets can influence the Senegal sole (*Solea senegalensis*) innate immune system [35]. Those authors have reported that stressed fish showed an increase in plasma lysozyme and alternative complement pathway activities as well as in the level of expression of g-type lysozyme (gLYS). In turbot (*Scophthalmus maximus*), dietary Arg could have enhanced some aspects of the stressed fish’s immune system, such as an increase in the relative percentage of circulating lymphocytes and peripheral monocytes, probably due to a higher availability of polyamines (derived from Arg) for leukocyte growth and differentiation [34].

Taurine (Tau; 2-aminoethanesulfonic acid), the most abundant free amino acid in marine animals, is associated with many physiological functions, such as the innate immune response [74]. It has been reported that Tau dietary supplements can protect tiger puffer fish (*Takifugu rubripes*) against environmental stress by enhancing the plasma non-specific immunity (complement C4 concentration) [54].

Methionine (Met) is an essential amino acid involved in the lipid metabolism [75]. Yang et al. [36] have studied the effects of dietary methionine (Met) on growth and other physiological responses in black seabream (*Acanthopagrus schlegelii*) submitted to nutritional stress (high-fat diets), concluding that Met inclusion could alleviate hepatic inflammatory response and apoptosis.

Gamma-aminobutyric acid (GABA) is a natural amino acid present in bacteria, plants, and animals and is synthesized from glutamic acid inside presynaptic synapses of inhibitory nerve cells [76,77]. Although it is not an alpha amino acid, it has been used as a feed additive in some fish species [78,79,80]. In fact, Bae et al. [62] have reported that GABA supplements improved macrophage maturation, autophagy activation, and antibacterial response to bacterial infection and enhanced the host’s innate immune response in olive flounder (*Paralichthys olivaceus*) under high stocking density stress.

### 3.2. Effects Involving Energy Metabolism

The secondary stress response leads to diverse metabolic changes, mainly those related to intermediate metabolism, in order to cope with a stressful condition and recover homeostasis, since stress is an energy-demanding process [49]. Several amino acids can act as energy substrates in those metabolic pathways; hence, their roles during the stress response can be crucial, affecting plasma stress markers related to energy mobilization as lactate and glucose [49]. For instance, amino acid metabolism involves aminotransferase reactions, which are stress markers in many animals, including humans [81,82]. Additionally, some amino acids can be converted into hormones and other substances that act directly on the energy metabolism [9]. Although it seems that amino acid supplementations usually provide beneficial metabolic effects, the addition of amino acids to the diet in an unbalanced manner can negatively affect growth parameters, as the requirements for other amino acids may be displaced by the supplemented one [75].

Tryptophan (Trp) is a neurotransmitter precursor (see previous sections), though 95% of ingested Trp is catabolized through the kynurenine pathway, producing niacin, pyruvate, and acetyl-CoA as the final products [67]. Acetyl-CoA and pyruvate can then enter into the Krebs cycle for making energy. In this sense, stress conditions induce a hypermetabolic status, which may induce the mobilization of amino acids as a coping mechanism for the increased energy demand, and Teixeira et al. [42] have reported that hepatic amino acid catabolism enzyme activity due to stress was affected by dietary Trp, decreasing with the increased level of Trp. Nevertheless, Herrera et al. [49] reported high gluconeogenic and glycolytic activities in stressed cods (*Gadus morhua*) fed Trp supplements. Those metabolic alterations can affect circulating plasma metabolites, and some works have reported that dietary Trp supplementations reduce plasma glucose and lactate levels in several fish species after stress submission [41,43,44,49,52]. However, Salamanca et al. [50] did not describe variations in glucose and lactate throughout the experimental period, which could be attributed to the long feeding duration with that amino acid. This aligns with the findings of Herrera et al. [18], who reported a decrease in plasma glucose and lactate concentrations when Trp levels were increased, but not when the feeding duration was prolonged. The metabolic changes under stress lead to growth alterations, and Trp supplements can reduce some growth parameters since they alter the food intake [42,44,48,50]. However, Tejpal et al. [41] described an improvement in growth parameters in *Cirrhinus mirgala* fed Trp supplements and subjected to high stocking density stress.

Phe is an essential amino acid that is metabolized through two metabolic pathways, namely oxidation to tyrosine (Tyr) and transamination to phenylpyruvate [83], and Tyr could be catabolized to hydroxyphenyl pyruvate and become a part of the energy metabolism [84]. It has been reported that both amino acids are mobilized to the brain as a stress response, resulting in a higher accumulation in fish fed Phe or Tyr supplements [57]. Additionally, Phe supplements kept or reduced enzyme activities related to the energy metabolism in the cod (*Gadus morhua*) [49]. Regarding plasma metabolites, it has been reported that Phe reduces glucose and lactate levels in specimens subjected to stress [49]. This is consistent with Salamanca et al. [57], who found a reduction in glucose and lactate levels in specimens fed diets supplemented with Phe and Tyr, except after 90 days of feeding. These results indicate that the effects of amino acids vary depending on the feeding duration and species. In previous studies, lactate levels were reduced in gilthead seabream and meager fed Phe and Tyr, but the reduction was more pronounced in meager [57]. The effects of those amino acids on growth have been studied in the gilthead seabream (*Sparus aurata*), describing that those supplements could decrease the growth parameters due to the unbalanced formulation (amino acid excess) provided on a long-term basis [57].

As methionine (Met) is involved in lipid metabolism (see previous section), its role in the intermediate metabolism can be crucial, mainly in stress conditions due to unbalanced feeding [85,86]. In this sense, Yang et al. [36] have reported that Met-supplemented diets can promote lipogenesis and suppress lipolysis, consequently reducing excess lipid accumulation and alleviating liver damage in stressed black sea breams (*Acanthopagrus schlegelii*) due to high-fat diets.

Valine (Val) and isoleucine (Ile) are branched-chain amino acids involved in several biochemical reactions, such as glycolysis and ketogenesis [87,88], key pathways in energy metabolism. Regarding stress response effects, a high level of Ile combined with a high level of Val in the diet improved the tolerance against low salinity stress in the Japanese flounder [61].

Taurine (Tau; 2-aminoethanesulfonic acid) regulates heart function and metabolism [89,90]. Therefore, Dixon et al. [91] have demonstrated that reducing cardiac taurine levels (through feeding) significantly affects whole animal sensitivity to acute thermal and hypoxic stress, impairs heart function, and influences energy metabolism in the brook char (*Salvelinus fontinalis*). However, dietary Tau did not reduce the plasma glucose concentration in *Mylopharyngodon piceus* and *Arapaima gigas*, though improved growth parameters [37,55].

GABA (gamma-aminobutyric acid) induces stable metabolic processes and immune and antioxidant responses by inhibiting overactive neurons in stressful situations [62]. In fish, it has been reported that dietary GABA had a beneficial effect on growth performance and feed utilization [80,92,93]. Bae et al. [62] did not find differences in GOT (aspartate aminotransferase) and GPT (alanine aminotransferase) activities in the olive flounder (*Paralichthys olivaceus*) between control fish and fish submitted to stress density and fed GABA additives; hence, they stated that was a good indicator of the beneficial effects of GABA as a feed additive.

Alanine–glutamine (Ala-Gln) is a dipeptide composed of the amino acids alanine and glutamine, both involved in various metabolic functions, such as protein synthesis and acid–base balance regulation [55,59]. Despite its role in cellular homeostasis and energy metabolism, it has been reported that Ala-Gln does not reduce glucose levels in the organism, regardless of its concentration in the diet [37,59]. This suggests that, although its supplementation may have other physiological benefits, it does not exert a modulatory effect on glucose under stress conditions or in basal situations.

Aspartate (Asp) has been identified as a regulator of glucose metabolism, acting as an intermediary in the Krebs cycle and gluconeogenesis [43]. Its impact on glucose levels varies depending on the type of stress the organism is exposed to, indicating that its effect is dependent on the physiological and environmental context. Studies have shown that Asp can play a key role in metabolic adaptation to adverse conditions, adjusting energy availability according to the organism’s demands [43].

### 3.3. Endocrine and Neuroendocrine Processes

Stress in fish induces significant hormonal changes, primarily affecting thyroid hormones, cortisol, and catecholamines. The hypothalamic–pituitary–interrenal (HPI) axis activation results in cortisol release, impacting immune functions and metabolic processes. Thyroid hormones regulate growth and metabolism, while catecholamines mediate acute stress responses. Several amino acids have been studied for their role in modulating these hormonal changes [44,69].

Tryptophan is a serotonin precursor, a neurotransmitter involved in stress response and regulation of the HPI axis. It is hypothesized that dietary Trp modulates stress through serotonergic activity, though exact mechanisms remain unclear. Studies indicate that Trp does not significantly reduce cortisol levels [43,52,68]. However, its supplementation may influence thyroid hormones and catecholamine levels.

Phenylalanine and tyrosine are precursors for catecholamines, impacting stress responses and metabolic functions. In gilthead seabream, Phe and Tyr supplementation led to decreased catecholamine levels, suggesting an influence on stress regulation [84]. However, in meager, these effects were not observed. Regarding cortisol, no significant differences were noted between stressed specimens supplemented with Phe and Tyr compared to controls [57,84]. Nonetheless, Herrera et al. [18] found that Phe supplementation influenced cortisol levels under air exposure stress but not thermal stress. Additionally, Tyr supplementation was associated with increased T4 (thyroxine) levels in meager [84].

Ala-Gln supplementation has been associated with increased T3 (triiodothyronine) and T4 levels, suggesting a stimulatory effect on thyroid function. This effect can be explained by the influence of glutamine on energy metabolism and hormone synthesis. Glutamine, as a key substrate for gluconeogenesis and the Krebs cycle, supports ATP production, providing the necessary energy for thyroid gland activity. Additionally, alanine participates in the glucose–alanine cycle, contributing to metabolic homeostasis and enhancing the availability of precursors for hormone synthesis [59].

Aspartate has not been shown to significantly reduce cortisol levels in stressed fish [43]. However, its role in energy metabolism suggests potential indirect effects on stress adaptation. By participating in the Krebs cycle and gluconeogenesis, aspartate contributes to ATP production, providing the necessary energy to maintain metabolic homeostasis and support the organism’s adaptive response to stressful conditions [94].

### 3.4. Responses Related to Oxidative Stress

Reactive oxygen species (ROS) are produced mainly in mitochondria under physiological conditions. When the level of ROS cannot be controlled and its content rises, oxidative stress develops [56]. Antioxidant enzymes, such as superoxide dismutase (SOD), catalase (CAT), glutathione peroxidase (GPx), and glutathione (GSH), are produced by cells to counteract oxidative stress by reducing ROS levels.

In aquaculture, numerous situations cause physiological stresses, including oxidative stress. The inclusion of certain amino acids in the diet can mitigate oxidative stress, significantly reducing ROS levels and decreasing the activity of antioxidant enzymes [36,56].

Several studies have shown that tryptophan, an essential amino acid precursor for serotonin and implicated in several behavioral patterns such as fear, stress [45], aggression [39], appetite regulation [35], immune response [47], social dominance, and sex behavior, could be a useful tool for reducing stress associated with farming practices [18,44,50]. An experiment investigating stress mitigation effects of dietary tryptophan (L-TRP) under thermal stress in rohu, *Labeo rohita* [53], showed that CAT and SOD activities were significantly higher in the control groups (0% L-TRP), whereas decreasing activities of these enzymes were observed with the increasing level of dietary L-TRP. It could be concluded that L-TRP might have antioxidative properties in mitigating oxidative stress caused by increased temperature.

Specific amino acids can scavenge ROS, and their dietary inclusion can alleviate oxidative stress. This is the case for methionine (Met) and taurine (Tau). Appropriate dietary levels of methionine in black seabream (*Acanthopagrus schlegelii*) alleviated physiological stress produced by a high-fat diet, reducing ROS levels while notably up-regulating the activity of antioxidant enzymes [36]. A similar situation was observed in European seabass *(Dicentrarchus labrax*) under sustained swimming conditions. A diet supplemented with 1.5% Tau resulted in significantly lower ROS levels than in a control diet, along with a significant decrease in the gene expression of antioxidant enzymes [56].

Leucine (Leu) is a functional amino acid that can be oxidized to supply energy. A recent study where Leu was added to the diet of sub-adult grass carp [60] showed that this AA could mitigate the negative effects of nitrite-induced stress response in this species. This may be attributed to leucine alleviating mitochondrial dysfunction by reducing ROS production.

## 4. Conclusions

Amino acid supplements have a significant influence on the stress response in fish. However, their long-term effects remain unclear and could potentially be negative due to the risks of unbalanced feeding. Despite this, their application prior to stressful conditions may offer benefits for fish welfare. Most studies have focused on essential amino acids such as tryptophan, phenylalanine, and tyrosine, although others, such as aspartate and taurine, have also been investigated. Since many amino acids are involved in both endocrine and neuroendocrine responses, as well as energy supply, understanding the primary pathways through which specific amino acids affect the stress system should be a key focus of future research.

## Figures and Tables

**Figure 1 animals-15-01293-f001:**
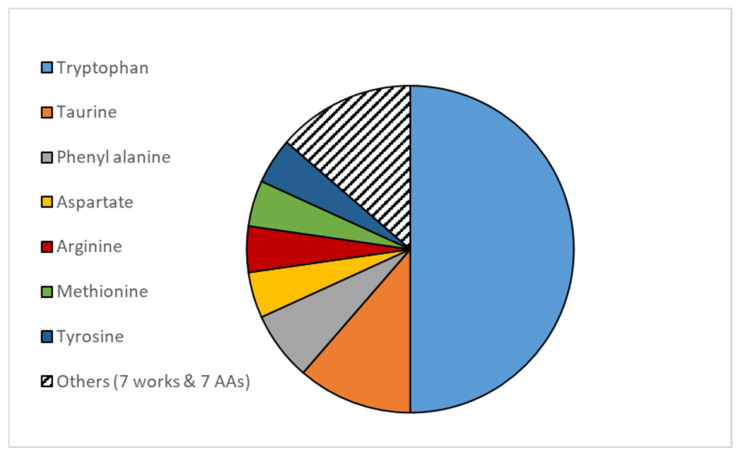
Distribution of papers dealing with the effects of amino acids on fish stress depending on amino acid type.

**Table 1 animals-15-01293-t001:** Relevant stress responses related to every amino acid found in the literature.

Amino Acid	Relevant Physiological Stress Responses	Species	References
Tryptophan	Improves feeding intake	*Salmo trutta*	Höglund et al. [35]
	Reduces aggressiveness without affecting stress indicators	*Brycon amazonicus*	Wolkers et al. [39]
	Decreases growth Increases food consumption	*Oncorhynchus mykiss*	Papoutsoglou et al. [40]
	Increases growth parameters under stress conditions Decreases cortisol and glucose levels Decreases energy requirements	*Cirrhinus mrigala*	Tejpal et al. [41]
	Decrease cortisol, glucose levels and energy requirements	*Argyrosomus regius*	Teixeira et al. [42]
	Alters the prl and gh expressions after stress Increases Hsp70 expression	*Argyrosomus regius*	Herrera et al. [43]
	Decreases enzyme activities related to amino acid and carbohydrate metabolism Increases the liver kynurenine concentration	*Argyrosomus regius*	Herrera et al. [44]
	Decreases plasma lactate and mucus glucose	*Argyrosomus regius*	Fernández-Alacid et al. [45]
	Upregulates immune-related gene expressions	*Argyrosomus regius*	Asencio-Alcudia et al. [46]
	Keeps levels of protease, antiprotease, peroxidase and lysozyme unchanged	*Argyrosomus regius*	Gonzalez-Silvera et al. [47]
	Reduces SOD and CAT activities Increases serotonergic activity and plasma cortisol	*Totoaba macdonaldi*	Cabanillas-Gámez et al. [48]
	Decreased plasma cortisol levels Increased liver transaminase activity Raises enzyme activity in glycolysis and gluconeogenesis	*Gadus morhua*	Herrera et al. [49]
	Reduces plasma cortisol and lactate	*Solea senegalensis*	Salamanca et al. [50]
	Increases plasma cortisol Modulates plasma glucose and lactate	*Solea senegalensis*	Herrera et al. [18]
	Increases plasma cortisol Higher brain monoamine content	*Dicentrarchus labrax*	Azeredo et al. [51]
	Increases plasma cortisol Decreases plasma glucose Decreases lysozyme and ACH50 Decreases serum thyroid hormone Inhibits post-stress immunosuppression Decrease serum thyroid hormones.	*Acipenser persicus*	Hoseini et al. [52]
	Decreases aminotransferase and lactate dehydrogenase activities Reduces enzyme activities related to oxidative stress Higher growth, RGR and PER	*Labeo rohita*	Kumar et al. [53]
Taurine	Improves growth performance, muscle composition and amino acid composition	*Takifugu rubripes*	Shi et al. [54]
	Enhances growth performance Improves intestine structure	*Mylopharyngodon piceus*	Tian et al. [55]
	Increases CAT activity Increases total serum immunoglobulin concentration	*Arapaima gigas*	Souto et al. [37]
	Increases growth Decreases ROS production and antioxidant enzyme gene expressions	*Dicentrarchus labrax*	Ceccotti et al. [56]
Phenylalanine	Lower plasma cortisol levels Increases liver transaminase activities Raises enzyme activity in glycolysis and gluconeogenesis	*Gadus morhua*	Herrera et al. [49]
	Reduces plasma stress markers	*Sparus aurata*	Salamanca et al. [57]
	Reduces plasma glucose and lactate	*Argyrosomus regius*	Salamanca et al. [57]
Aspartate	Enhances pomc-a expression Increases Hsp70 expression	*Argyrosomus regius*	Herrera et al. [43]
	Produces over-exudation of mucus metabolites and cortisol	*Argyrosomus regius*	Fernández-Alacid et al. [45]
	Keeps protease, antiprotease, peroxidase and lysozyme levels stable	*Argyrosomus regius*	Gonzalez-Silvera et al. [47]
Methionine	Decreases TG, TC, NEFA, LDL-C, and ALT Decreases lipid droplets in liver Increased ampkα and sirt1 expression Improves lipogenesis pathway gene expressions Up-regulates antioxidant enzyme activities and gene expression levels Decreases pro-inflammation and pro-apoptosis gene expressions Up-regulates anti-inflammatory cytokine and anti-apoptosis gene expressions	*Acanthopagrus schlegelii*	Yang et al. [36]
	Increases plasma cortisol Upregulates complement factor 3 Increases immune cells	*Dicentrarchus labrax*	Azeredo et al. [51]
Tyrosine	Reduces plasma stress markers	*Sparus aurata*	Salamanca et al. [57]
Arginine	Decreases plasma cortisol levels	*Scophthalmus maximus*	Costas et al. [34]
	Increases respiratory burst activity and nitric oxide production of head kidney leukocytes Enhances HIF-1, HAMP-1, MIP1-alpha and gLYS expressions	*Solea senegalensis*	Costas et al. [58]
Alanine + Glutamine	Increases body weight Increases fish survival during a bacterial challenge	*Cyprinus carpio*	Chen et al. [59]
Leucine	Promotes FW, WG, PWG, and SGR Decreases activities of serum parameters Decreases ROS, NO and ONOO^−^ activities Increases mRNA levels of mitochondrial biogenesis genes and fusion-related genes Decreases mRNA levels of fission-related genes, mitophagy-related genes and autophagy-related genes	*Ctenopharyngodon idella*	Zhen et al. [60]
Valine + Isoleucine	Enhances growth Increases blood parameter levels	*Paralichthys olivaceus*	Shi et al. [61]
Gamma-aminobutyric acid	Improves macrophage maturation, autophagy activation, and antibacterial response to bacterial infection	*Paralichthys olivaceus*	Bae et al. [62]

Hsp70: mitochondrial heat shock protein 70; SOD: superoxide dismutase; CAT: catalase; ACH50: alternative complement pathway; RGR: relative growth rate; PER: Protein efficiency ratio; ROS: oxygen-containing reactive species; pomc-a: proopiomelanocortin-derived hormone; TG: triglyceride; TC: total cholesterol; NEFA: non-esterified fatty acid; LDL-C: low-density lipoprotein cholesterol; ALT: alanine transaminase; HIF-1: hypoxia-inducible factor 1; HAMP-1: hepcidin antimicrobial peptide 1; MIP1-alpha: macrophage inflammatory protein 1α; gLYS: g-type lysozyme; FW: final weight; WG: weight gain; PWG: percent weight gain; SGR: specific growth rate; NO: nitric oxide; ONOO^−^: peroxynitrite.

## Data Availability

Not applicable.

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
