# Peer review of "Amino Acids as Dietary Additives for Enhancing Fish Welfare in Aquaculture"

_animals, 2025, doi:10.3390/ani15091293_

Round 1
Reviewer 1 Report
Comments and Suggestions for Authors
General comments: The manuscript titled "Amino acids as dietary additives for enhancing fish welfare in aquaculture" provides a comprehensive review of the role of amino acids in mitigating stress and improving welfare in aquaculture fish. The topic is highly relevant given the increasing focus on animal welfare in aquaculture and the potential of dietary interventions to enhance fish health and productivity. The authors have done a thorough job of summarizing the existing literature, particularly in relation to the physiological and metabolic effects of various amino acids on fish under stress conditions. However, there are several areas where the manuscript could be improved to enhance its clarity, depth, and overall impact. The manuscript would benefit from a more thorough proofreading to correct minor grammatical errors and improve the overall flow of the text.
Specific comments:
- The introduction provides a good overview of the importance of fish welfare in aquaculture and the role of stress in affecting fish health. However, it would benefit from a more detailed discussion of the specific challenges faced in aquaculture that lead to stress, such as overcrowding, poor water quality, and handling practices. This would provide a stronger context for the subsequent discussion on amino acids.The authors should also consider briefly mentioning the economic implications of improved fish welfare, as this could strengthen the rationale for the study.
- Line 44, what is “IMTAs”? the readers may not know.
- Section 2, it would be helpful to include a brief discussion on the differences in amino acid metabolism between fish and mammals, as this is crucial for understanding the specific needs of fish in aquaculture.
- Line 65, amino acids (AAs). So do the others.
- Line73, incomplete expression of “ “essential” (EAA)”.
- Table 1,the text that follows could be more concise and focused. Some of the information is repetitive, and the section could benefit from a more structured approach, perhaps by grouping amino acids with similar effects together. Besides, the second column of the table may not be inappropriate in terms of the relationship between “feeding intake or growth performance” and physiological stress responses. Of note, most of the literature is outdated, and we also recommend the use of literature for representative species, such as carnivorous and phytophagous fishes.
- Section 3, the authors should consider adding a brief discussion on the potential interactions between different amino acids when used in combination. This is an area that is not well-covered in the current literature but could be of significant interest to researchers and practitioners.
- Section 4-7, the discussion on the mechanisms by which amino acids influence immune responses could be expanded. For example, the authors could delve deeper into the immune modulation, metabolic pathways, hormonal changes, specific antioxidant enzymes involved by specific amino acids, etc....Besides, the authors should also consider discussing the potential for amino acids to enhance vaccine efficacy and influence gut microbiota and reproductive hormones in fish, as this is an emerging area of research with practical implications for aquaculture.
- Conclusions, the authors should consider adding a brief discussion on the potential for future research in this area,highlighting the need for more studies on the interactions between different amino acids and the potential for amino acids to enhance the efficacy of other dietary interventions.
Author Response
1. The introduction provides a good overview of the importance of fish welfare in aquaculture and the role of stress in affecting fish health. However, it would benefit from a more detailed discussion of the specific challenges faced in aquaculture that lead to stress, such as overcrowding, poor water quality, and handling practices. This would provide a stronger context for the subsequent discussion on amino acids. The authors should also consider briefly mentioning the economic implications of improved fish welfare, as this could strengthen the rationale for the study.
The Introduction has been modified according to this comment.
2. Line 44, what is “IMTAs”? the readers may not know.
The definition has been specified.
3. Section 2, it would be helpful to include a brief discussion on the differences in amino acid metabolism between fish and mammals, as this is crucial for understanding the specific needs of fish in aquaculture.
We take your suggestion into account, and it has been added to the text.
4. Line 65, amino acids (AAs). So do the others.
It has been corrected in the text.
5. Line73, incomplete expression of “ “essential” (EAA)”.
It has been corrected in the text.
6. Table 1, the text that follows could be more concise and focused. Some of the information is repetitive, and the section could benefit from a more structured approach, perhaps by grouping amino acids with similar effects together. Besides, the second column of the table may not be inappropriate in terms of the relationship between “feeding intake or growth performance” and physiological stress responses. Of note, most of the literature is outdated, and we also recommend the use of literature for representative species, such as carnivorous and phytophagous fishes.
Thank you for your observation. It is important to note that Table 1 is a summary of the main findings reported in the literature on amino acids in aquaculture-relevant species. Although some of the literature may appear outdated, it was included because it corresponds to key studies conducted on this topic in species of interest for aquaculture.
7. Section 3, the authors should consider adding a brief discussion on the potential interactions between different amino acids when used in combination. This is an area that is not well-covered in the current literature but could be of significant interest to researchers and practitioners.
The effects of amino acids have been considered individually and not the interaction between two amino acids, as this is, as you rightly mention, an area that is not widely studied.
8. Section 4-7, the discussion on the mechanisms by which amino acids influence immune responses could be expanded. For example, the authors could delve deeper into the immune modulation, metabolic pathways, hormonal changes, specific antioxidant enzymes involved by specific amino acids, etc....Besides, the authors should also consider discussing the potential for amino acids to enhance vaccine efficacy and influence gut microbiota and reproductive hormones in fish, as this is an emerging area of research with practical implications for aquaculture.
The effects on metabolism, antioxidant enzymes and endocrine/neuroendocrine system are explained in other sections. However, this section on immune responses has been modified according to this comment. Regards vaccines, those are not considered in this paper because it is focused on amino acids used as feeding additives.
9. Conclusions, the authors should consider adding a brief discussion on the potential for future research in this area,highlighting the need for more studies on the interactions between different amino acids and the potential for amino acids to enhance the efficacy of other dietary interventions.
It has been added to the text.
Reviewer 2 Report
Comments and Suggestions for Authors
The inclusion of beneficial additives in the diet in order to mitigate the stress response to typical stressors in aquaculture has been an important research topic, and within it, the use of amino acids. In this review, the main roles of amino acids in fish stress were summarized. The topic is pertinent and given the amount of work available, it is welcome a review like this. MS is well organized, written andis worthy to be published.
A couple of minor comments:
- Revise caption for Table 1. Symbols of genes should be deleted.
- Given that the MS is on the application of amino acid supplements in aquafeeds, it sounds obligated a section (not necessarily separated) on the feasibility to supplement commercial diets with amino acids. Is this already done commercially? Which method is more suitable for achieving this? Direct supplementation with crystalline amino acids? Use of particular protein sources or practical ingredients such as hydrolysates, etc? How much this would impact feed cost?
Author Response
The inclusion of beneficial additives in the diet in order to mitigate the stress response to typical stressors in aquaculture has been an important research topic, and within it, the use of amino acids. In this review, the main roles of amino acids in fish stress were summarized. The topic is pertinent and given the amount of work available, it is welcome a review like this. MS is well organized, written andis worthy to be published.
A couple of minor comments:
• Revise caption for Table 1. Symbols of genes should be deleted.
It has been corrected
• Given that the MS is on the application of amino acid supplements in aquafeeds, it sounds obligated a section (not necessarily separated) on the feasibility to supplement commercial diets with amino acids. Is this already done commercially? Which method is more suitable for achieving this? Direct supplementation with crystalline amino acids? Use of particular protein sources or practical ingredients such as hydrolysates, etc? How much this would impact feed cost?
Thanks for your comment. This review basically deals with physiological effects of dietary AAs and is not focused on potential sources or feed making. However the Introduction has been modified according to this comment.